# Chromosome Transplantation: Opportunities and Limitations

**DOI:** 10.3390/cells13080666

**Published:** 2024-04-11

**Authors:** Angela La Grua, Ilaria Rao, Lucia Susani, Franco Lucchini, Elena Raimondi, Paolo Vezzoni, Marianna Paulis

**Affiliations:** 1Department of Medical Biotechnologies and Translational Medicine, University of Milan, 20129 Milan, Italy; angela.lagrua@humanitasresearch.it; 2IRCCS Humanitas Research Hospital, 20089 Rozzano, Italy; ilaria.rao@humanitasresearch.it (I.R.); lucia.susani@irgb.cnr.it (L.S.); 3Department of Biomedical Sciences, Humanitas University, 20072 Pieve Emanuele, Italy; 4UOS Milan Unit, Istituto di Ricerca Genetica e Biomedica (IRGB), CNR, 20138 Milan, Italy; 5Department for Sustainable Food Process, Università Cattolica del Sacro Cuore, 29122 Piacenza, Italy; franco.lucchini@unicatt.it; 6Department of Biology and Biotechnology “L. Spallanzani”, University of Pavia, 27100 Pavia, Italy; elena.raimondi@unipv.it

**Keywords:** chromosome transplantation, genomic disease, X chromosome, Duchenne muscular dystrophy, iPSC

## Abstract

There are thousands of rare genetic diseases that could be treated with classical gene therapy strategies such as the addition of the defective gene via viral or non-viral delivery or by direct gene editing. However, several genetic defects are too complex for these approaches. These “genomic mutations” include aneuploidies, intra and inter chromosomal rearrangements, large deletions, or inversion and copy number variations. Chromosome transplantation (CT) refers to the precise substitution of an endogenous chromosome with an exogenous one. By the addition of an exogenous chromosome and the concomitant elimination of the endogenous one, every genetic defect, irrespective of its nature, could be resolved. In the current review, we analyze the state of the art of this technique and discuss its possible application to human pathology. CT might not be limited to the treatment of human diseases. By working on sex chromosomes, we showed that female cells can be obtained from male cells, since chromosome-transplanted cells can lose either sex chromosome, giving rise to 46,XY or 46,XX diploid cells, a modification that could be exploited to obtain female gametes from male cells. Moreover, CT could be used in veterinary biology, since entire chromosomes containing an advantageous locus could be transferred to animals of zootechnical interest without altering their specific genetic background and the need for long and complex interbreeding. CT could also be useful to rescue extinct species if only male cells were available. Finally, the generation of “synthetic” cells could be achieved by repeated CT into a recipient cell. CT is an additional tool for genetic modification of mammalian cells.

## 1. Introduction

Although the genes responsible for thousands of monogenic diseases have been identified, for the majority of them, therapeutic options remain limited. Satisfactory results have recently been obtained in some hematological or neurological rare monogenic disorders due to small abnormalities that can be corrected by delivery of the entire transcript or by gene editing with several kinds of processing nucleases [1,2]. However, there are conditions arising from chromosomal aberrations characterized by numerical and structural defects of the genome, such as aneuploidies, large deletions or inversions, complex intrachromosomal rearrangements, and copy number variations in which these approaches might be difficult to implement. These defects could be defined as “genomic mutations” since they involve large segments of the genome, sometimes entire chromosomes [3]. In theory, most of these mutations could potentially be corrected through chromosome transplantation (CT). We define CT as the precise substitution of an endogenous chromosome harboring such kinds of mutations with a normal exogenous one, resulting in the formation of a genetically corrected diploid cell.

Historically, chromosome transfer in mammalian cells can be traced back to hybrid cell construction. Highly proliferating neoplastic cells were fused with donor cells and selected for the presence of specific chromosomes. In addition to the production of monoclonal antibodies, these hybrid cells were used to tackle several biological problems. During the first phase of the Human Genome Project, interspecific hybrids containing limited portions of the human genome have been instrumental in mapping genes into specific chromosome portions and isolating genes responsible for human diseases.

Subsequently, the Microcell-Mediated Chromosome Transfer (MMCT) technique enabled the efficient transfer of individual chromosomes through the production of microcells containing a single chromosome [4,5]. This technique was later used to transfer chromosomal vectors, namely mammalian and human artificial chromosomes (MACs, HACs) [6,7,8]. MACs and HACs have been successfully produced via two approaches: (1) the top-down approach, which consists of the progressive reduction of a natural chromosome to a minimal one [9,10,11], and (2) the bottom-up approach, in which cloned centromeric satellite DNA, telomeric DNA and genomic DNA are co-transfected into host cells where new mini-chromosomes are originated [12]. More recently, bottom-up approaches to create fully synthetic mammalian artificial chromosomes made significant advances [13,14,15,16,17]. Both MACs and HACs applications range from basic research to applied research and include the construction of models to investigate the biology of chromosome functional elements (first of all, the centromere) [18] and the construction of transchromosomic animals as models for human diseases [19], one of the greatest challenges being the application of HACs in gene therapy [20]. However, the use of artificial chromosomes as shuttles for gene transfer implies the addition of the MAC/HAC as a supernumerary element to a normal karyotype.

Notably, no attempt has been made to reconstruct normal diploid cells using this method. Actually, until recently, the accurate replacement of an endogenous chromosome with an exogenous one had not been pursued. This was partly due to the requirement of normal proliferating cells to achieve this objective. The recent availability of embryonic stem cells (ESCs) and induced pluripotent stem cells (iPSCs), characterized by a normal diploid genome, indefinite growth, pluripotency and capacity to differentiate into various tissues, has opened new avenues for utilizing CT in correcting genetic anomalies.

## 2. The Technique of CT

CT can be accomplished in three steps (Figure 1).

The first one consists of the choice of a recipient and a donor cell line. For the reasons listed above, iPSCs are natural candidates for the role of the recipient cells. The choice of a donor cell is usually based on a neoplastic heterologous cell which has a propensity to micronucleate and to produce microcells. After the isolation of a donor cell clone stably maintaining the desired chromosome, the donor cell is ready to be used. In the second step, prolonged exposure of the donor cells to an inhibitor of tubulin polymerization such as colchicine leads to the accumulation of metaphase cells. During this process, the nuclear membrane reorganizes around single chromosomes or small groups of chromosomes, resulting in the formation of “micronuclei”.

Subsequently, cells can be “micro-enucleated” by centrifugation in the presence of a microfilament-disrupting agent such as cytochalasin B. This process yields microcells, composed of single-chromosome-containing micronuclei enclosed by the plasma membrane [4]. Next, these microcells are fused with the appropriate recipient cells according to the MMCT protocol, followed by selection based on an endogenous gene or an antibiotic selection cassette. In both cases, cell clones containing the desired single chromosome are isolated after selection and confirmed by cytogenetic analysis. In the third step, cells maintaining the exogenous chromosome but spontaneously losing the endogenous defective one are isolated. Alternatively, the elimination of the endogenous defective one could be facilitated by various approaches (see below).
Figure 1Schematic overview of the CT approach: the diagram illustrates the three main steps. 1. Generation of recipient male (XY) iPSCs containing a defective X chromosome (depicted in red), and a donor cell line (LA9 or CHO) capable of forming microcells, containing a normal X chromosome (depicted in green). 2. Fusion between microcells and recipient X-defective cells, followed by the selection of XXY cells. 3. Elimination of the extra sex chromosome (either Y or defective X), resulting in the generation of a normal CT euploid XX or XY iPSC line.
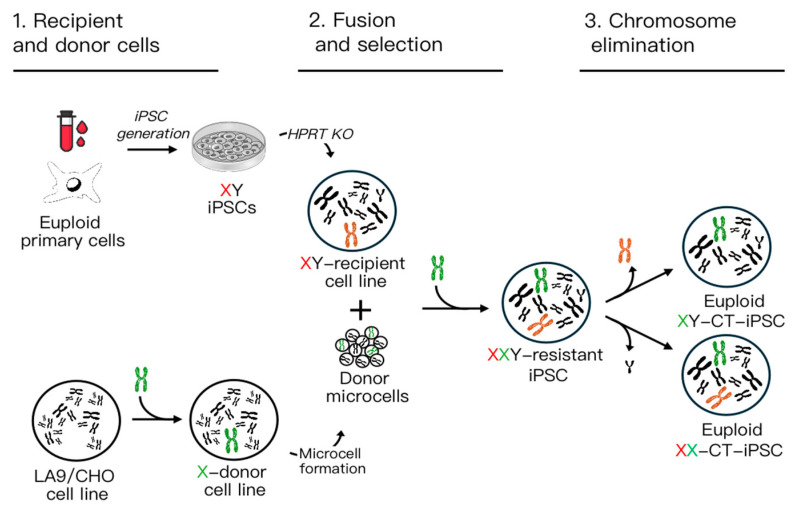



### 2.1. Step 1: Donor Cell Line and Micronucleation

As mentioned above, the donor cells must possess the ability to produce microcells. The efficiency of micronucleation and microcell formation is closely related to the cell type [21]. For instance, inducing micronucleation in spontaneously immortalized mouse LA9 fibroblasts or Chinese Hamster Ovary (CHO) cells is notably simpler compared to primary cell lines like fibroblasts. Achieving micronucleation requires actively proliferating cells. Adjusting both the colchicine concentration and the exposure duration is crucial for different cell lines. Micronucleation assessment can be easily conducted through phase-contrast microscopic examination. We developed a CHO hybrid cell line containing a normal human X chromosome derived from normal fibroblasts that could be used for the correction of iPSCs derived from any patient affected by X-linked “structural” disorders. Therefore, this cell line can be regarded as a “universal X chromosome donor” [22].

A strategy aiming at improving the production of single-chromosome-containing microcells from the donor cell lines could target the micronucleation step. It is conceivable that if more microcells are produced, the rate of chromosome transfer could be enhanced. The classical protocol makes use of colcemid, a microtubule destabilizer, followed by disarrangement of the actin cytoskeleton using Cytochalasin B, to promote microcell formation. Recently, Uno and coworkers have shown that both microcell production and chromosome transfer efficiency can be improved by substituting colcemid with an association of taxol (paclitaxel) and reversine [23]. Taxol is a microtubule stabilizer while reversine is an inhibitor of the spindle assembly checkpoint. The combination of 400 nM Taxol and 500–1500 nM reversine was shown to be superior to colcemid and also to the combination of two mitotic spindle disruptors (TN-16 and griseofulvin) in several cell lines and with different fusogens [23]. In addition, Liskovykh and coworkers showed that the substitution of Cytochalasin B with Latrunculin B significantly increased (*p* < 0.0001) the efficiency of MMCT [24]. Although these data were obtained by performing transfer of artificial chromosomes, there is no reason why the method should not work with entire human chromosomes; moreover, it might permit CT in cell lines other than iPSCs, such as human primary cells.

### 2.2. Step 2: Fusion and Selection

To improve the efficiency of CT in human iPSCs it could be possible to exploit a more efficient fusogenic activity. The conventional MMCT method, utilizing Polyethylene Glycol (PEG), has long been employed to induce fusion between microcells and target cells for chromosome transfer [25]. However, PEG’s cytotoxicity and low fusion efficiency pose significant challenges, particularly for sensitive cell types like primary cells and stem cells. In response, researchers have explored alternative fusogens mainly based on modified viral proteins, which act as fusogens [26,27].

Initially, we established a hybrid LA9 mouse cell line (LA9/human) incorporating various human chromosomes, including the X chromosome (Homo sapiens X chromosome [HSAX]). Unfortunately, we were unable to obtain stable CT clones which acquired the exogenous X chromosome from this hybrid cell line. This difficulty arose from the low efficiency of the MMCT technique performed in iPSCs. Faced with these challenges, we decided to adopt a more recently described strategy known as retro-MMCT (Figure 2).

This innovative approach takes advantage of the fusogenic properties of the R peptide-deleted envelope (EnvΔR) protein derived from the amphotropic murine leukemia virus (MLV) [28]. To implement the retro-MMCT system, it is crucial to consider that the Env protein utilized in this approach is fusogenic for both human and mouse cells, since both express the receptor for the Env protein. Consequently, after Env infection, human or murine cells fuse, forming syncytia. This made our previously generated LA9/human cell line unsuitable for retro-MMCT. To address this limitation, we explored a system to generate a non auto-fusogenic donor cell line containing the desired HSAX for transplantation. To achieve this goal, we opted for a hypoxanthine-guanine phosphoribosyltransferase (Hprt)-defective CHO cell line capable of forming microcells without expressing the receptor for the Env protein. Subsequently, we introduced a single HSAX derived from our LA9/human cell line into the CHO genome using MMCT. Then, we selected resistant CHO/HSAX hybrid clones. Confirmation of the presence of HSAX was achieved through fluorescence in situ hybridization (FISH) using a whole-X-chromosome-painting probe. In theory, employing a similar approach opens the possibility of generating chromosome donor cell lines tailored to each autosome.

All these approaches require the infection of donor cells with a vector containing the fusogen before the fusion step. One of these alternatives, the HVJ-E fusion technique, which utilizes inactivated the Hemagglutinating virus of the Japan-derived envelope protein, exhibited lower cytotoxicity compared to traditional PEG methods [26]. The MV-MMCT approach, employing Measles virus envelope proteins, demonstrated improved efficiency and limited toxicity, albeit limited to human-derived recipient cells due to viral tropism [27]. The recent approach of retro-MMCT, exploiting murine leukemia virus envelope proteins, presents a substantial leap forward, significantly increasing chromosome transfer efficiency across various mammalian cells. This improvement holds promise for applications in diverse fields, including CT.

Within this methodology, a normal human X chromosome can be transferred from a hybrid rodent donor cell line to the recipient iPSCs by utilizing the established strategy of *HPRT* gene-based selection. The normal donor X chromosome is HPRT-positive (HPRT^+^), while the defective X chromosome in the recipient cells must lack HPRT functionality.

It is noteworthy that HPRT^+^ cells can be selected in the HAT (Hypoxanthine, Aminopterin, and Thymidine) medium, while defective HPRT cells can be selected in 6-thioguanine (6-TG) medium. Since the *HPRT* gene in patient-derived iPSCs is wild type (with the exception of those derived from Lesch–Nyhan patients), it must be inactivated in cell lines of interest by transfection with an *HPRT*-targeting Cas9- construct containing the Cas9 nuclease and a specific single guide RNA (sgRNA). Subsequently, cells are exposed to the 6-TG medium to isolate the HPRT-defective cells.

Then, HPRT defective iPSC clones can be fused with microcells obtained by micronucleation of the normal HSAX-containing donor cell (CHO/HSAX). Following the retro-MMCT procedure, which involves infecting CHO/HSAX cells with lentivirus particles encoding the EnvΔR protein, the fusion between microcells and recipient HPRT-defective iPSCs is promoted. This system, initially developed for mouse cells and adapted to human iPSCs in our laboratory, enhances the efficiency of chromosome transfer [22].

After fusion, cells undergo selection in the HAT medium. Clones capable of growing in HAT must have acquired the normal HSAX from the donor CHO/HSAX cells. Selection is maintained for one or two weeks, and resistant clones are expanded and independently analyzed.

The HPRT-based selection offers a straightforward and efficient approach to correct X-linked diseases. In diseases involving autosomes, alternative selection systems warrant consideration. An ideal choice would be opting for endogenous selectable genes, as it ensures the absence of any residual exogenous sequences in the final corrected cells. However, only a limited number of genes in the genome, such as that based on the endogenous thymidine kinase (TK) located on the human chromosome 17 [29], possess this distinctive characteristic. If no endogenous system is exploitable, it is possible to insert a classical antibiotic-based selection cassette which could be excised by the Cre-LoxP system after successful CT. Nevertheless, this approach does not ensure the absence of residual exogenous sequences in the final corrected cells.

### 2.3. Step 3: Chromosome Elimination

In vivo, non-neoplastic aneuploid cells are genomically quite stable, although the existence of mosaic patients is well known. In these cases, it is assumed that aneuploid cells are prone to lose the extra chromosome probably because of a growth disadvantage. In the last few years, it became clear that this phenomenon can also occur in vitro, where the rescue of a diploid state can be easily detected and isolated. This has been the basis of the isolation of rescued diploid clones in both mouse and human cells [22,30,31], which can be achieved by cytogenetic examination or PCR screening (for example, it is possible to identify the presence of the wild type and the absence of the mutated *HPRT* gene corresponding to the exogenous and endogenous X chromosome, respectively). Recently, Murakami and coworkers (see below) exploited the natural tendency to lose the Y chromosome to generate 39,X0 mouse iPSCs as a first step to produce 40,XX normal female cells from male cells [32].

However, the rescue of diploid cells is an uncontrolled event, since in our hands, it cannot be predicted or even stimulated. For this reason, a number of methods to eliminate the extra chromosomes have been suggested (reviewed in [33]). The simplest one, already used by Capecchi in his seminal work on knockout mouse generation, is based on the insertion on the to-be-eliminated chromosome of a thymidine kinase (TK) gene derived from the herpes simplex virus (HSV) which allows the incorporation of the cell DNA of a nucleotide analog which is toxic for the cell: in this case, cells which have not lost the HSV-TK-bearing chromosome are selected against [34,35,36].

Another strategy uses the Cre-LoxP system, which requires the insertion of a cassette in the targeting chromosome and a subsequent transfection with the Cre recombinase [35,37,38]. This method carries the risk of chromosome fragmentation and reinsertion.

A simpler approach is based on CRISPR technology, exploiting the cutting activity of the system by multiple targeting the desired chromosome either with an sgRNA directed to a repeated sequence or with different sgRNAs targeting single copy sequences; in this way, the digested chromosome is eliminated [39,40]. However, there is the risk of off-target activity and chromosome fragmentation and reinsertion, although careful analysis of the obtained clones can identify “*bona fide*” diploid clones.

More recently, an approach based on the targeting of alpha satellite repeats of 19 out of 24 human centromeres by defective Cas (dCas9) fused to a mutant KLN1 (KaryoCreate) has been reported to be able to create both losses and gains on ten human chromosomes [41]. Tovini and coworkers used a dCas9-CENP-T fusion protein to create new centromeres to cause aneuploidies, including chromosome loss [42], while Truong et al. used dCas9 to link microtubule minus-end-directed Kinesin 14VIb to subtelomeric (HSA1) or centromeric (HSA9) repeated sequences to interfere with mitosis, thus causing the loss of the targeted chromosome [43]. Similarly, Girish and coworkers reported the elimination of chromosome 1 from a neoplastic cell line by modification of the HSV-TK and CRISPR strategies [44]. However, all these strategies were tested on neoplastic cells and their utility on normal pluripotent cells has not been investigated. As a matter of fact, their effect on genome stability was not investigated due to the neoplastic nature of the cells used. 

In conclusion, as far as pluripotent normal stem cells are concerned, we believe that the simple approach of screening a reasonable number of spontaneous clones after fusion remains the best choice to maintain genome integrity. The HSV-TK or a CRISPR-based strategy could be used in selected cases if the desired clones had not been identified.

## 3. Results Achieved So Far

Some years ago, we set up a project to investigate whether a CT approach could be useful to treat genetic diseases. We chose the X chromosome as a model for several reasons. First, due to its hemizygote state in males, a disproportionate number of human diseases map to this chromosome. Second, we reasoned that if the elimination of the endogenous chromosome would prove difficult, a therapy could still be possible with the 47,XXY cells, since an additional X chromosome is easily tolerated in human cells due to X inactivation. Third, a viable diploid cell would be produced also if the Y sex chromosome was lost instead of the defective endogenous X one.

Until recently, no attempt was made to genetically modify structural abnormalities. In 2012, two groups investigated genetic engineering approaches to eliminate the trisomic chromosome 21 in iPSCs from patients with Down’s syndrome [36,45]. Surprisingly, it was found that diploid cells that had lost the extra chromosome 21 were present in the cultures. In 2014, Bershteyn and coworkers showed that in iPSCs derived from a patient with single-ring chromosome 17, a high percentage of cells spontaneously rescued the defect simply by losing the defective chromosome and doubling the correct one: cells had a diploid content, although chromosome 17 was isodisomic [46,47].

These findings paralleled what we had found in mouse cells. We started with Hprt-defective mouse ESCs as recipient cells and performed MMCT with a hybrid cell line containing a normal mouse X chromosome as a donor cell. The efficiency of the procedure was very low, but in repeated experiments, we were able to obtain several 41,XXY clones, which rapidly lost either the endogenous X or the Y chromosome, giving rise to 40,XY or 40,XX clones. The procedure neither affected the genome stability of the clones nor their ability to differentiate. The CT 40,XX cells were injected into blastocysts and several chimeric pups were born, containing both XY and XX cells (if the original blastocyst was XY), although we were unable to obtain germline transmission [30]. These cells were also used as nuclear donors in cloning experiments but we were unable to obtain live mice. 

The *HPRT* mutation in human patients with the Lesch–Nyhan syndrome allows the direct selection of CT cells via the procedure described above. However, for all the other genetic diseases mapped to the X chromosome, the implementation of this kind of selection needs the inactivation of the endogenous *HPRT* gene on the affected chromosome. This can be achieved by simple classical CRISPR technology which is usually highly efficient when used to inactivate genes. Therefore, to show the general applicability of the CT strategy to every X chromosome genetic defect, we chose a mouse model of human chronic granulomatous disease due to mutations in the *CYBB* gene which maps to the X chromosome both in humans and mice [48]. In humans, a high proportion of patients affected by this defect bear large deletions which are a good candidate to be cured by CT [49]. In addition, after CT, iPSCs could be forced to differentiate toward the granulocyte lineage and the resulting cells could be injected in patients. Indeed, it has already been demonstrated that these patients can benefit from granulocyte infusion from unrelated donors [50]. We were able to show that, after *Hprt* gene inactivation in iPSCs derived from a mouse model of this disease and fusion with a donor cell line, normal diploid clones, able to differentiate to functionally corrected granulocytes, could be easily obtained [31]. In this way, a theoretically infinite number of autologous granulocytes could be produced. This strongly suggested that the defect of any gene located on the X chromosome could be treated in this way.

Next, we moved to show that CT could be performed in human iPSCs. An HPRT-deficient iPSC line was reprogrammed from a patient affected by Lesch–Nyhan syndrome and processed by retro-MMCT. We selected 47,XXY and both diploid 46,XX and 46,XY were obtained, which were able to differentiate to all the three embryonic germ layers and showed genomic stability. Exome sequencing confirmed that no relevant mutation occurred during the CT procedure and SNP analysis confirmed that in diploid cells, the X chromosome was of donor origin [22].

Further studies have shown that by the CRISPR-mediated *HPRT* gene inactivation approach used in mice [31], human-chromosome-transplanted iPSCs can be obtained from patients affected by other diseases mapping to the X chromosome such as Duchenne muscular dystrophy (DMD). DMD is one of the best candidates for CT since its gene is larger than 2 Mb and it is often due to gross deletions that cannot be treated by conventional gene therapy. Although the defect in vitro has been corrected by HAC [51,52], this approach was only partially effective in a large animal such as the pig [53]. As a matter of fact, despite the various therapeutic approaches to gene therapy that have been proposed, none of them can fully resolve the defect of gross mutations for which the current treatments only convert DMD into a less severe phenotype [54,55,56,57].

## 4. CT for Autosomes

So far, data on CT are available only for the X chromosome. Whether this technique will be applicable to autosomes is a matter of speculation. However, the fact that cultured iPSCs are prone to spontaneously lose a trisomic chromosome, which is an obligatory intermediate in CT, suggests that at least the third step of the procedure would not be limiting. Working with iPSCs from human autosomal trisomies, several groups have found that diploid cells can be spontaneously recovered from the cultures. This has been shown for trisomy 21 [36,45], for trisomies 18, 13 and 9 [33], for trisomy 1 [44] and for trisomy 16 [32]. In addition, even if a few human chromosomes were resistant to spontaneous elimination, the techniques reviewed above for engineered chromosome elimination could be used.

Likewise, step one, including the establishment of iPSCs from patients and the preparation of donor cells with single human chromosomes, can be simply accomplished. MMCT should also be easily performed, but the selection system needs to be evaluated. For X chromosome transplantation, the *HPRT* gene provides a very simple selection system, but cannot be used for autosomes. Notably, CT on the X chromosome restores a perfect diploid genome at the end of the procedure without any scarring left behind, since the defective HPRT-inactivated chromosome is replaced with a completely normal one. A few endogenous systems located on autosomes, like endogenous TK, have been described and could be used. Alternatively, if the genetic defect involves a molecule that could be easily detected by immunological means, CT clones could be identified by FACS analysis and sorting, without any genetic modification. Finally, it would be possible to use standard selection cassettes inserted in “safe harbors” of the donor chromosome, which could be verified by sequencing. In this case, the selection marker would remain in the genome but in a location where it should not give any risk.

## 5. Male to Female

When we initially investigated different approaches to the elimination of the additional sex chromosomes, we realized that by simply culturing 47,XXY cells, diploid clones, which had lost either chromosome, sometimes occur [22,30]. Apparently, this event occurs independently from the fact that the additional X chromosome was added by CT or mis-segregated during meiosis, since both female and male cells were recently obtained from iPSCs from Klinefelter patients [58]. Although this possibility is not a rare event taking into consideration all our CT experiments both in mice and humans, we were unable to find any clue that could explain why cells lose the Y or the X chromosome. So far, this choice is unpredictable, although it has long been known that losing the Y chromosome is a frequent occurrence in long cultured cell lines. Therefore, apparently, the possibility of obtaining both female and male cells from the same male iPSC line with CT is not a rare event (Figure 3) [22,30].

This possibility could be exploited in cases in which, for various reasons, it could be interesting to perform this modification. Recently, the Hayashi group, who pioneered the generation of gamete cells from mouse pluripotent stem cells, showed that female cells could be obtained by a two-step procedure in which isolation of male clones, which had spontaneously lost the Y chromosome by in vitro culture, was the first step [32]. Losing the Y chromosome from ESCs or iPSCs during culture is a relatively frequent event and Eggan and coworkers previously exploited this finding by isolating 40,XY and 39,X0 clones that were then used to create mice by tetraploid complementation, thus obtaining both female and male progeny of the same parental ESC/iPSC clone [59]. This was possible because in some mouse strains, 39,X0 females are fertile. However, in humans, 46,X0 females are usually not fertile (Turner syndrome); hence, it is likely that this approach would require the addition of an exogenous X chromosome to be used in humans. Based on the same 39,X0 generation approach, Deng and coworkers produced live mouse pups whose genome was derived from two (different) male cells by a quite complex procedure in which 39,X0 clones, isolated from a mouse germline competent 40,XY pluripotent stem cell line, were used to generate, by blastocyst injection, female pups which were then bred to normal males, thus obtaining mice derived from two fathers [60].

Murakami and collaborators rescued the diploid content of 39,X0 clones by implementing a second step in which cells were exposed to reversine, a drug that, by inhibiting a spindle checkpoint, enhances chromosome mis-segregation. Clones in which the two X chromosomes were retained in the same cell could be isolated, thus giving rise to 40,XX cells, which were further differentiated into functional female gametes [32]. The fact that the two X chromosomes were identical (uniparental disomy) apparently did not affect oocyte development, although it might cause recessive genetic diseases in the newborn if a mutation was present in the original “duplicated” chromosome. In this regard, we believe that the CT procedure could be a simpler and perhaps safer alternative to the generation of 46,XX females from human male cells. A comment on the ethical problems associated with the use of CT for genetic sex change in humans can be found in a recent paper by Sparrow [61].

Although performed essentially in mouse and human cells, it is likely that the CT approach to generate XX cells could be feasible also in other mammals. Suvà and coworkers cloned horses from 64,XY cells obtaining several male horses and an adult 63,X0 female; as expected, however, the animal had small underdeveloped ovaries, suggesting gonadal hypoplasia [62]. This approach opens the possibility of obtaining female cells which could be used in endangered mammalian species in which only male animals or even only male cells are available. Obviously, a female surrogate species is needed for embryo transfer and the procedure must be adapted to each species, which makes this possibility quite remote, especially because the physiology of the reproduction of endangered species is usually poorly investigated. Finally, the use of the described procedures in humans to obtain oocytes from male cells for only-male couples would be theoretically possible, but ethical and/or technical reasons so far preclude this kind of experiment [61,63].

## 6. Large Animals

Somatic cloning and CRISPR-Cas technology are the most recent breakthroughs in the genetic engineering of large animals, while QTL (quantitative trait loci) mapping and marker-assisted selection are commonly employed in animal breeding [64,65,66]. Several QTL have been identified in livestock, which are linked to productivity, disease resistance and/or adaptation to hostile environments. It could be possible to breed different strains of livestock with different advantageous QTL but, for large animals, this is a long and costly procedure. Similarly, besides canonical productive traits, breeders are currently aiming to reintroduce traits from wild populations to withstand the heat-stress and reduce production losses induced by climate change. When genome editing is not an option, because large genomic modifications are required, chromosome transfer could be a favorable alternative to a full cross-breeding, allowing a faster isolation of the desired genotype. It might be possible, for instance, to introduce a disease resistant QTL-containing chromosome into iPSCs from a strain characterized by high productivity which can subsequently be used for cloning, thus saving time and costs. In perspective, CT could become an additional instrument in the toolbox of the modern breeder. However, so far this possibility remains speculative.

## 7. Limitations

Despite its potentiality, the CT approach has important limitations (Table 1).

First of all, it cannot be applied directly in vivo, but the strategy must be performed in vitro on highly proliferative cells. In the last few years, there has been a shift in gene therapy toward the in vivo procedure which is not feasible for CT. An improvement in the efficiency of the procedure could perhaps permit in the future to perform CT in hematological stem cells ex vivo. This means that CT could be applied just in patient cells able to grow in vitro and then reinjected after correction.

So far, CT has been performed only with the X chromosome and it is not clear whether it would be feasible with all the other 22 human autosomes. However, the fact that diploid cells could be obtained for some aneuploidies and ring chromosomes, it is conceivable that at least for some (perhaps the smallest) chromosomes the CT could be successful. On the other hand, once set up, the same donor cell line and the procedure could be used for every disease mapping on a specific chromosome.

Another limitation is shared with all the therapeutic approaches based on cultured cells for long periods, especially reprogrammed cells. Although the MMCT procedure itself does not seem to cause additional points or small mutations, alterations with oncogenic potential could occur during reprogramming and/or extensive culture. However, since only one or a few iPSC clone(s) are sufficient to produce differentiated cells, it is quite possible to perform whole genome sequencing in several CT clones and use only the best ones for therapeutic applications.

## 8. Conclusions

This review examines the state of the art of the CT technique and discusses its potential applications that extend beyond the treatment of human diseases (Figure 4).

Our research demonstrates the feasibility of obtaining female cells from male cells through CT. This manipulation could present an opportunity for generating female gametes from male cells. Furthermore, CT holds potential in veterinary biology by transferring entire chromosomes carrying advantageous loci to animals of commercial interest. It could also aid in the preservation of extinct species if only male cells are available. Finally, the generation of “synthetic” cells could be achieved through repeated CT. With successive cycles of CT, each involving a distinct chromosome, the possibility of substituting each chromosome emerges, paving the way for the creation of entirely artificial cells. In conclusion, chromosome transplantation represents a valuable tool for the genetic modification of mammalian cells, opening new horizons in biotechnology.

## Figures and Tables

**Figure 2 cells-13-00666-f002:**
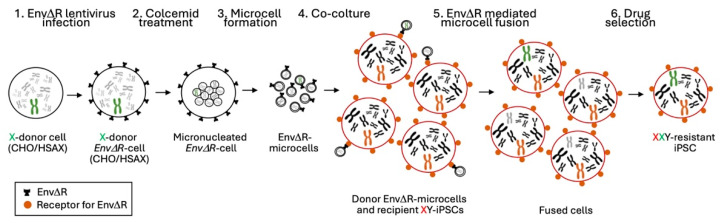
Schematic overview of the retro-MMCT approach: the diagram illustrates the main steps. 1. A donor cell line containing a normal human X chromosome (CHO/HSAX) is infected with the EnvΔR lentivirus. 2–3. Colcemid treatment induces micronucleation and after centrifugation microcells from the donor cells are obtained; 4–5. Microcells are co-cultured with the recipient iPSCs to induce EnvΔR-mediated cell–microcell fusion. 6. The resulting fused cells are selected to identify those in which the normal HSAX has been acquired.

**Figure 3 cells-13-00666-f003:**
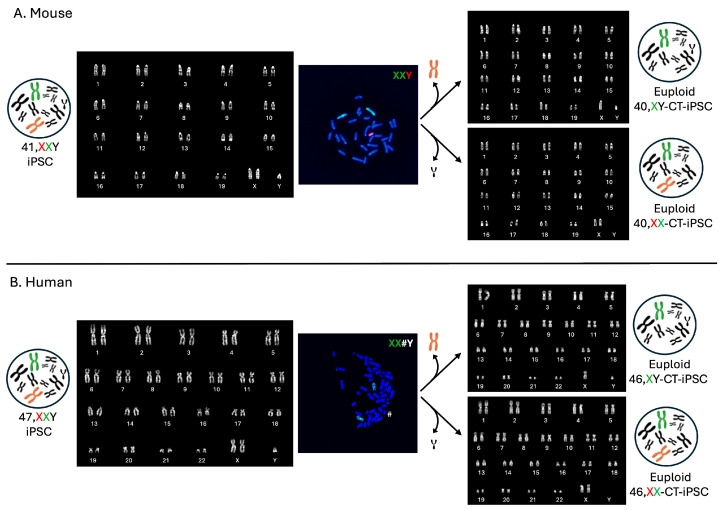
Representative images illustrate the occurrence of spontaneous in vitro loss of an extra sex chromosome. We isolated cells that spontaneously lost either the additional X or the Y chromosome in both mouse (**A**) and human (**B**) cells. (**A**) Representative karyotype (left) and metaphase spread (middle) following FISH with probes for the mouse X chromosome (green) and Y chromosome (red) are presented. On the right are two representative karyotypes of cells obtained after a few passages in culture from the original 41,XXY, showing the presence of cells that have spontaneously lost the extra sex chromosome (40,XY or 40,XX). (**B**) Representative karyotype (left) and metaphase spread (middle) following FISH with a probe for the human X chromosome (green) are shown. The human Y chromosome (#) is identified by banding. On the right are two representative karyotypes of cells obtained after a few passages in culture from the original 47,XXY, indicating the presence of cells that have spontaneously lost the extra sex chromosome (46,XY or 46,XX).

**Figure 4 cells-13-00666-f004:**
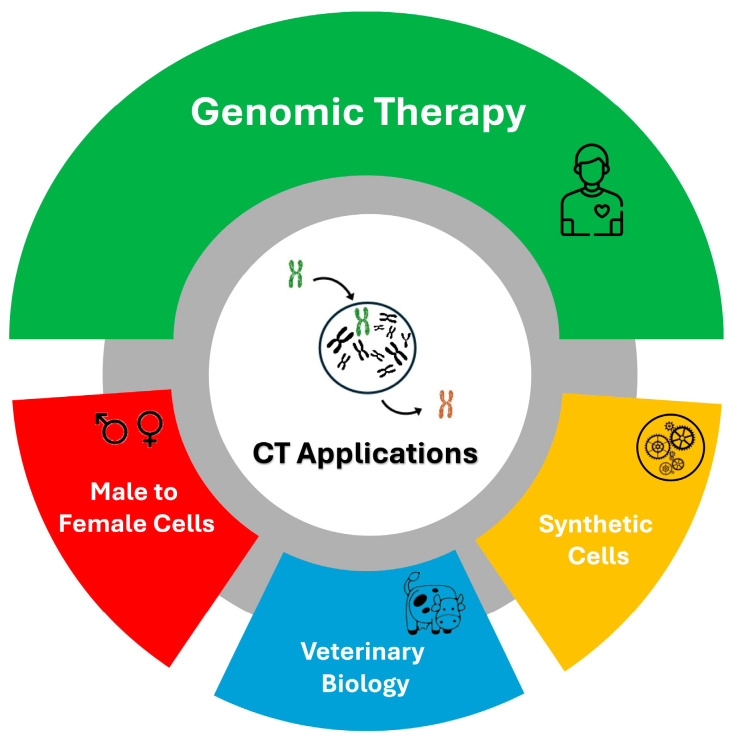
Possible applications of CT.

**Table 1 cells-13-00666-t001:** Achievements and limitations of the CT approach.

	Achievements	Limitations
Donor cell line (CHO ^1^/HSA ^2^)	Single-chromosome universal donor cell line for the treatment of the disorder mapping to the specific chromosome. Cell line able to form microcells.	Demonstrated only for X-linked disorders (CHO/HSAX).
Recipient cell line (iPSCs ^3^)	Cells with a normal diploid genome, indefinite growth, pluripotency and capacity to differentiate into various tissues.	Long term in vitro culturing could accumulate mutations.
Fusion (retro-MMCT ^4^)	Retro-MMCT is based on murine leukemia virus envelope protein (Env) mediated fusion: less cytotoxic and more efficient compared to classic PEG ^5^ based MMCT fusion.	It is necessary to infect the donor cells with a lentivirus carrying the Env gene.
Selection (drug selection)	Use of an endogenous selectable gene (i.e., *HPRT* ^6^) or classical drug selection.	Endogenous selection system limited for a few chromosomes.
Chromosome loss (spontaneous loss)	Spontaneous loss of trisomic chromosomes without genomic manipulation.	Alternatively, more complex approaches could be used to drive the chromosome loss.

^1^ CHO: Chinese Hamster Ovary cell line; ^2^ HSA: Homo sapiens chromosome; ^3^ iPSCs: induced pluripotent stem cells; ^4^ MMCT: microcell-mediated chromosome transfer; ^5^ PEG: Polyethylene Glycol; ^6^
*HPRT*: Hypoxanthine-guanine phosphoribosyltransferase.

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
