# Peer review of "Chromosome Transplantation: Opportunities and Limitations"

_cells, 2024, doi:10.3390/cells13080666_

Round 1

Reviewer 1 Report

Comments and Suggestions for Authors

Comments for the Authors:

In this manuscript, La Grua et al. reviewed the technological process and possible applications of Chromosome Transplantation (CT) technique. This review systematically introduces the three-step technique of CT and its achievement. Moreover, possible developments of CT such as genome therapy, synthesized cells with exogenous chromosomes were imagined. In the end, the limitations existing in this technique were discussed. La Grua et al. provided insights into the potential therapeutic applications of CT technique.

Major comments:

1. For part 2.2, it’s better to add a figure for a more understandable description of the retro-MMCT approach.

2. For part 5, as mentioned in this part, CT approach can obtain female cells which only male animals are available. This description is biased because there are species like chicken using ZW type gender determination system, and CT cannot obtain female cells from male cells in these species. So, additional explains need to be added.

Minor comments:

pp.5 line 176: “significantly” should be followed with P-value or citations.

pp. 5 line 212 "the Cre/LoxP system" and pp. 6 line 236 "the Cre-LoxP system": unify two terms.

pp.9 line 385: ”ad” should be “and”.

There are many abbreviations in the manuscript that do not appear twice such as ’HVJ’ and ‘CGD’, which are unnecessary.

Author Response

We thank the reviewer for her/his suggestions. 

Major comments:

  1. For part 2.2, it’s better to add a figure for a more understandable description of the retro-MMCT approach.

In part 2.2 we added a figure (now, Figure 2) to better describe the Retro-MMCT approach.

  1. For part 5, as mentioned in this part, CT approach can obtain female cells which only male animals are available. This description is biased because there are species like chicken using ZW type gender determination system, and CT cannot obtain female cells from male cells in these species. So, additional explains need to be added.

We agree that in part 5 we cannot use the term "animals" because birds and other organisms have a different kind of sex determination. Therefore, we substituted the term "animals" with "mammals".

Minor points:

pp.5 line 176: “significantly” should be followed with P-value or citations.

We added the significance level (Now, pp.4 line. 153)

pp. 5 line 212 "the Cre/LoxP system" and pp. 6 line 236 "the Cre-LoxP system": unify two terms.

We unified the Cre-LoxP system 

pp.9 line 385: ”ad” should be “and”.

We corrected the word "and"

There are many abbreviations in the manuscript that do not appear twice such as ’HVJ’ and ‘CGD’, which are unnecessary.

We eliminated the abbreviations that do not appear twice

Plese see the attachment with the revised manuscript

Reviewer 2 Report

Comments and Suggestions for Authors

Dear authors and editor,

Thank you for the invitation to review this manuscript. In this article, the authors discuss chromosome transplantation, presenting in the review the advances of this technique, as well as possible areas that would benefit and the limitations for applying the technique. The authors have experience in the area as can be seen from previous publications and therefore have knowledge on the subject. The text is easy to understand and contains relevant information. However, at some points the text is very repetitive. Ex. Several points presented in item "3. Results Achieved so Far" are presented in other points in the text, so I suggest that it be revised so as not to repeat information. Furthermore, I suggest that a table be created presenting the main information for each step of the technique with the limitations and advances already achieved.

I recommend publication with minor changes.

Author Response

We thank the reviewer for her/his pertinent suggestions.

We eliminated a number of repetitions in parts 2.2 and 3 and added a Table (Table 1) highlighting both achievements and limitations of the CT technique, as suggested by the referee.

Please see the attachment with the revised manuscript